# Gender and Neurosteroids: Implications for Brain Function, Neuroplasticity and Rehabilitation

**DOI:** 10.3390/ijms24054758

**Published:** 2023-03-01

**Authors:** Loredana Raciti, Caterina Formica, Gianfranco Raciti, Angelo Quartarone, Rocco Salvatore Calabrò

**Affiliations:** IRCCS Centro Neurolesi Bonino Pulejo, 98121 Messina, Italy

**Keywords:** neurosteroids, neuroplasticity, excitability, GABA-receptors, estrogen, gender, hippocampus, hypothalamus, rehabilitation

## Abstract

Neurosteroids are synthesized de novo in the nervous system; they mainly moderate neuronal excitability, and reach target cells via the extracellular pathway. The synthesis of neurosteroids occurs in peripheral tissues such as gonads tissues, liver, and skin; then, because of their high lipophilia, they cross the blood–brain barrier and are stored in the brain structure. Neurosteroidogenesis occurs in brain regions such as the cortex, hippocampus, and amygdala by enzymes necessary for the in situ synthesis of progesterone from cholesterol. Neurosteroids could be considered the main players in both sexual steroid-induced hippocampal synaptic plasticity and normal transmission in the hippocampus. Moreover, they show a double function of increasing spine density and enhancing long term potentiation, and have been related to the memory-enhancing effects of sexual steroids. Estrogen and progesterone affect neuronal plasticity differently in males and females, especially regarding changes in the structure and function of neurons in different regions of the brain. Estradiol administration in postmenopausal women allowed for improving cognitive performance, and the combination with aerobic motor exercise seems to enhance this effect. The paired association between rehabilitation and neurosteroids treatment could provide a boosting effect in order to promote neuroplasticity and therefore functional recovery in neurological patients. The aim of this review is to investigate the mechanisms of action of neurosteroids as well as their sex-dependent differences in brain function and their role in neuroplasticity and rehabilitation.

## 1. Introduction

Neurosteroids (NSs) were named in 1981 [1] and identified as steroids that are synthesized de novo in the nervous system, such as in the hippocampus and other brain structures, and are accumulated in the nervous system autonomously from the steroidogenic endocrine glands. Neurosteroids have been implicated in several neurological mechanisms, such as epileptogenesis, hepatic encephalopathy, neurodegeneration, neuroprotection, and psychiatric disorders. NSs’ origins from cholesterol or other steroidal precursors derive from circulating steroid hormones. They mainly moderate neuronal excitability [2,3] and reach target cells via extracellular pathways. These paracrine signals modulate neurotransmitter-gated ion channels and G-protein-coupled receptors, predominantly [4,5] γ-aminobutyric (GABA)A2 and the N-methyl-D’Aspartate (NMDA) receptors [4,5]. Excitability is carried by chemicals signals discharged from astrocytes, oligodendrocytes, Schwann cells, and neurons such as Purkinje cells, hippocampal neurons, and retinal amacrine and ganglion cells of the brain [6,7]. 

Following the structural characteristics, NSs can be classified as:(i)pregnane, such as allopregnanolone (5α-pregnane-3α-ol-20-one) and allotetrahydrodeoxycorticosterone (THDOC, 5α-pregnane-3α,21-diol-20-one), whose precursors are progesterone and deoxycorticosterone, respectively;(ii)androstane, such as androstanediol and etiocholanone, derived from testosterone and estradiol;(iii)sulphated, such as pregnenolone sulfate (PS) and dehydroepiandrosterone sulfate (DHEAS).

The NSs cross the brain barrier and stimulate brain function. Neurons and glial cells display activity of 5α-Reductase [8,9], whereas neocortex and subcortical white matter, as well as hippocampal tissues, have 5α-reductase and 3α-HSOR enzymes activities [10,11].

Moreover, the brain astrocytes and glutamatergic principal neurons express cytochrome P450 cholesterol side-chain cleavage enzyme (CYP450scc) that transforms cholesterol to pregnenolone [12,13]. Therefore, a translocator protein (18 kD) present in the peripheral tissues and in the brain, acting as a peripheral or mitochondrial benzodiazepine receptor [14,15], supports the moving of cholesterol through to the inner mitochondrial membrane [16]. Successively, in the inner mitochondrial membrane the availability of cholesterol to the CYP450scc increases, so cholesterol is converted to pregnenolone, which is a key intermediate for NSs biosynthesis. Moreover, the presence of the 3β-hydroxysteroid dehydrogenase enzyme that converted the pregnenolone to progesterone has been shown in the brain [17] (Figure 1).

Neurosteroidogenesis occurs in the brain regions such as the cortex, hippocampus, and amygdala by enzymes necessary for the in situ synthesis of progesterone from cholesterol [18,19]. NSs may act via multiple pathways, by regulating gene transcription or through a direct action on neurotransmitter-gated ion channel receptors and G-protein-coupled receptors. The most important targets are the γ-aminobutyric (GABA)A [20] and the NMDA receptors [21,22,23,24]. Nevertheless, the biosynthesis of NSs in the brain is still unclear. 

The aim of this review is to investigate the mechanisms of action of the NSs, their sex-dependent differences in brain function, and their role in neuroplasticity and rehabilitation. 

## 2. Mechanism of Action of Neurosteroids

There are two chronic effects of NSs in the brain: genomic (classical intracellular steroid receptors), due to their metabolic interconversion to traditional steroids [3], and non-genomic rapid actions, mediated by ion channels and membrane receptors.

The effects of NSs on the brain excitability modulation depend on the interaction between neural membrane receptors and ion channels with rapid effects (within minutes), in contrast to the slow effects of steroid hormone via intracellular steroid receptors (even though a prolonged duration has been shown) [22]. Moreover, it has been shown that NSs directly and widely control the GABA-A receptors, the ligand-gated ion channels [23], in positive or negative phase depending on the type of the steroid molecule [24].

Structurally, GABA-A receptors are heteropentamers with five protein subunits that form the chloride ion channels to moderate the bulk of synaptic inhibition in the central nervous system. NSs increase the chance of opening the GABA-A receptor chloride channel, given that the closed time increases the chloride current through the channel with a reduction of neuronal excitability. The GABA function includes the opening of chloride ion channels, and the internalization of the chloride ion to facilitate the hyperpolarization of the membrane [25,26,27]. GABA-A receptors bypass the depolarization of the excitatory neurotransmission and avoid the action potential generation by two types of inhibitory neurotransmission: synaptic (phasic) and extrasynaptic (tonic) inhibition, that are modulated and potentiated by NSs. The GABA-A receptor subunit influences the NSs action modulation of GABA-A receptors [23,24,25,26,27,28,29]. Specifically, the α-subunit influences the NSs efficacy, whereas the γ-subunit may control both the efficacy and potency [27]. The precise site of NSs binding is currently unknown, although it seems to be placed at highly conserved glutamine (position 241 in the M1 domain of the α-subunit), playing a key role in NSs modulation. Therefore, the effect of NSs is probable due to their action on both synaptic and perisynaptic/extrasynaptic GABAA receptors.

The intermittent release of high levels of GABA, from presynaptic axon terminals of GABAergic interneurons, triggers the γ2-containing receptors at the synapse, inducing the phasic/synaptic inhibition. The tonic/inhibition results from the extrasynaptic continuous activation of δ-containing receptors, related to low levels of ambient GABA molecules that escape reuptake by GABA transporters [27,28,29]. The δ-subunits are located perisynaptically/extrasynaptically and they mediate the “tonic” GABA-A receptor current [20,28,29,30,31], producing a stable inhibition of neurons and decreasing their excitability.

Furthermore, the location of the δ-subunit on the dendrites of hippocampal dentate gyrus granule cells supports the GABA-A receptors to work as a controller of hippocampus excitability.

Allopregnanolone (ALLO), THDOC, and androstanediol are potent positive allosteric modulators of GABA-A receptors [4,5,32], binding the D ring at both C20 of the pregnane steroid side chain and C17 of the androstane of the ring A with a positive activity at GABA-A receptors [19].

ALLO has anxiolytic, sedative–hypnotic, and anticonvulsant effects. Alcohol, hydroxybutyrate, and diazepam may potentiate GABAergic inhibition directly or through the increased availability of ALLO [20]. Pregnanolone sulphate (PS) and DHEAS are sulphated at C3 (called “sulphated steroids”) and block GABA-A receptors at low micromolar concentrations, reduced the channel opening frequency [24,33,34,35,36] and inhibited actions [24,36]. They are also effective allosteric agonists at the NMDA receptor complex and negative non-competitive modulators of the GABA-A receptor [37]. High micromolar concentrations of PS and DHEAS accomplish actions on NMDA receptor-mediated currents and act on the σ1 receptors with presynaptic action, inducing glutamate release. Therefore, NSs can also modulate the NMDA type glutamate receptors [38]. The NMDA receptors exhibit two distinct sites for NSs modulation: one facilitates the effects of positive modulators, while the other pleads the effects of negative modulators. The results may have implications on mechanisms of cognition, neuroprotection, and neurotoxicity [21]. Such receptors may modulate the release of acetylcholine and dopamine, neurotransmitter systems involved in memory, motor control and behavior [39]. PS increases the fractional open time of NMDA-activated channels, by increasing the frequency and the duration of the channel opening based on the subunit ligand, and inhibits the NMDA-induced [3H] norepinephrine. The NR2A and NR2B subunits supported a potentiating effect, while NR2C and NR2D subunits sustain an inhibitor effect [40]. DHEAS and PS, as well as pregnenolone, cooperate with σ1 receptors in the brain [40]. DHEAS and PS perform an agonist action, while progesterone acts as an antagonist.

On the other hand, DHEAS potentiate the NMDA-evoked excitability of hippocampus neurons, an effect that could be blocked by the σ1 antagonist haloperidol and NE-100, as well as by progesterone [40].

Therefore, whereas pregnenolone sulphate exercises excitotoxic effects on cortical and retinal cells, DHEA and DHEA-S have a neuroprotective effect against glutamate toxicity in vitro. The brain NSs vary in concentrations through time with different physiological mechanisms, depending on aging, stress, menstrual cycle, pregnancy, menopause and neurologic and psychiatric disorders [41]. Mameli et al. showed that NSs are implicated in correct brain development: some NSs may act as retrograde messengers, encouraging plasticity in immature synapses during development [42]. In fact, the administration of some positive modulators of GABAergic function such as diazepam, a benzodiazepine agonist, cause several variations in GABA-mediated functions in adulthood, with impaired locomotion and exploration as consequences [43]. Moreover, it has been shown that the enzymes necessary for neurosteroidogenesis are expressed in the immature brain and that the treatment with NSs of neuronal cells in vitro induces trophic effects, especially in the case of progesterone. The result was the boost of dendritic outgrowth in Purkinje cells. At the same time, the authors showed that ALLO helps the formation of neuronal circuitry that supported the persistence of neurons’ development. On the other hand, ALLO administered in rat pups caused an impairment of the diffusion of interneurons in the adult prefrontal cortex due to a physiological age fluctuation of ALLO in the rat brain [42]. This latter finding showed a first prenatal peak of cortical levels of ALLO and a second peak in the second week of life [44]. The administration of 10 mg/kg of ALLO on the fifth postnatal day caused an impairment of the localization and function of prefrontal and dorsal thalamic GABAergic neurons in the adult rat brain [44,45]. Moreover, perinatal NSs administration might modify the normal development of the hippocampus and the striatal and cortical dopaminergic activity [46,47]. 

Other research in rats by Darbra et al. showed that the administration of ALLO, during neonatal age, manipulates the behavioral affects in adolescents and adults [48]. 

In particular, the administration of finasteride to inhibit neonatal ALLO diminished the novelty exploratory behavior, decreased unspecific body weight and increased anxiety-related behavior during adolescence. Moreover, the neonatal administration of ALLO progressively declines the prepulse inhibition (PPI) of the acoustic startle response in adulthood, indicating an impairment of the sensorimotor gating [48,49]. The deficiency of sensorimotor gating is characteristic of various neuropsychiatric disorders, such as schizophrenia [50]. Furthermore, the effects of NSs on the dopaminergic system have also been investigated. Li et al. showed that finasteride, administrated during adolescence, inhibited the dopaminergic system in late adolescent male rats [51]. The impairment of the dopaminergic system caused the inhibition of exploratory and motor behaviors, and a drop in dopamine metabolites in the frontal cortex, hippocampus, caudate putamen, and nucleus accumbens of late adolescent male rats. The inhibition of the 5α reductase II of these areas caused a block of dihydrotestosterone production and, consequently, androgen production, with a lack of stimulation of the dopaminergic system. 

Moreover, a down-regulation of tyrosine hydroxylase mRNA and protein expressions in the substantia nigra and ventral tegmental area has been shown. No delayed dopaminergic effect has been demonstrated after the administration of finasteride during the first early post-natal period. This result suggested that the finasteride effect on the dopaminergic system is mediated by the inhibition of the activity of androgen. Consequently, androgen acts on central nervous system function [52,53], stimulating the activity of the dopaminergic system, and finasteride could be used as a therapeutic option for neuropsychiatric disorders such as schizophrenia or Tourette syndrome.

## 3. Gender Differences in NSs Action

Sex difference is one of the long-standing issues in neuroscience research concerning certain brain disorders.

Testosterone is either irreversibly converted to estradiol (E2) by the activity of aromatase or metabolized to DHT by the activity of 5α-reductase. The sexual NSs (SN) DHT and E2 concentrations in the hippocampus are significantly higher than in the serum of males and females, respectively [54]. E2 and DHT are synthesized de novo from cholesterol that is transported through the mitochondrial membranes by the steroidogenic acute regulatory protein (StAR), which is expressed in the hippocampus of male and female animals [55].

Ovariectomy and gonadectomization in males reduced the hippocampal dendritic spine density [56,57]. These mechanisms of action could be blocked by letrozole, an aromatase inhibitor, in females, as well as by finasteride, a 5α-reductase inhibitor, in males [58].

E2 enhances the cellular model for learning and memory in the hippocampus, the so-called long-term potentiation at the CA3-CA1 synapses, increases the number of NMDA receptor binding sites, without effect on AMPA receptor binding sites [55], and enhances the immunofluorescence of the NMDA receptor subunit NR1 in females. Therefore, the result is a block of the NR2B NMDA receptor subunit that abolishes E2-induced enhancement of LTP. Consequently, E2 induces the magnitude of LTP at the CA3-CA1 Schaffer collaterals in the hippocampus.

Fester et al. [58] showed that the receptors for Gonadotropin releasing hormone (GnRH) regulate E2 synthesis in the ovaries and, subsequently, SN synthesis in the hippocampus. Therefore, due to the estrous cycle, GnRH are released cyclically from the hypothalamus in females; thus, the brain SN levels are correlated to related to sex hormones and the reproductive state of the organism throughout life. On the other hand, after GnRH stimulation, similar effects in males were not shown [58] and it has been assumed that GnRH stimulates testosterone synthesis, which is converted to DHT in males. The double function of increasing spine density and enhancing LTP has been related to the memory-enhancing effects of sexual steroids. In ovariectomized animals, treatment with E2 and testosterone increased CA1 pyramidal spine synapse density [59]. The local release of E2 induces testosterone to the rescue of spine density [60]. On the other hand, in orchiectomized males, the only steroids that restore the orchiectomy-induced spine synapse loss are either testosterone or the non-aromatizable DHT. Therefore, as previously reported, it could be assumed that synaptic plasticity in the hippocampus is sex-dependent, with the E2 sex steroid in females and testosterone in males [61].

Moreover, Fester et al. [58] highlighted the role of SN as the principal player in SN, which induced hippocampal synaptic plasticity [61]. Additionally, these results provided evidence that the continuous synthesis of NSs is required for normal transmission in the hippocampus [55,58,61].

Treatment with E2 has led to a beneficial function of memory in women; meanwhile, androgens have shown positive results in working memory tests on male animals [61,62]. As we have seen, the mechanisms of action of NSs are different for gender. A study conducted in a healthy population showed a correlation between NSs such as DHEA, DHEA-S, and pregnenolone regarding cognitive function and psychological well-being differences between genders. In males, serum DHEA levels correlated positively with quality of life while DHEA-S and pregnenolone levels were correlated with cognitive function. On the other hand, a correlation between DHEA-S and working memory was found in females [62]. These results indicate that NSs had a relevant role in cognitive function and quality of life, with a difference between genders [63,64,65].

## 4. Neuroplasticity and the Role of Neurosteroids

Neuroplasticity is considered the capacity of neural networks to modify their structure and function in response to environmental and biological inputs. It is regulated by different mechanisms, but NSs seem to play an important role. NSs are found in the brain at high concentrations, and their presence after steroidogenesis suggests that there is a local synthesis [66,67]. However, to better understand the role of NSS in modulating brain function, it is noteworthy to talk about neuroplasticity in terms of neurogenesis, structural and functional plasticity [42,61]. In fact, the earliest work on neuroplasticity was conducted by Hebb, which assumed the involvement of structural and functional changes that seem to occur in excitatory (glutamate) and inhibitory (GABAergic) networks [68,69]. Structural plasticity refers to the dimension of the neural arbor, dendritic length and number or ramifications [69], while functional plasticity is considered as an increase or decrease of electrical activity in cerebral regions, called LTP and long-term depression (LTD), respectively [70]. LTP and LTD are sustained by NMDA and AMPA receptors and modulated by GABAergic neurotransmission [71]. Various studies have demonstrated that NSs have been characterized as neuroplasticity modulators, regulating neurogenesis and structural and functional plasticity [72]. Schverer et al. [72] described the effects that specific NSs had in neuroplasticity, regarding Pregnenolone (PREG), Dehydroepiandrosterone (DHEA), their sulphate derivatives, PREGS and DHEAS, progesterone (PROG), and ALLO [54]. PREG increases the functional plasticity through NMDA receptors and stimulates NMDA receptors in newborn neurons. DHEA increases functional plasticity in terms of synaptic efficacy developed LTP via NMDA receptor signaling, and increases short-term potentiation, neurogenesis and structural plasticity through an increase in spine density. ALLO potentiates both mature excitatory synapses in vitro, possibly via presynaptic GABA-receptors modulation, and neurogenesis through GABA receptor activation in neuroprogenitor cells [23,24]. It is known that steroids may act through a genomic action mediated by a specific steroid receptor, as well as through a nongenomic action mediated by certain receptors for neurotransmitters or neuromodulatory proteins [73,74,75]. 

Based on these assumptions, steroids are considered neuroactive steroids, they have pharmacological effects specifically on the receptor GABAA, NMDA [76] and on the sigma-1 receptor [74]. In fact, progesterone and some of its metabolites ALLO are potent positive modulators of the GABAergic function, while DHEA, PREG, and their sulfate esters are negative modulators of the GABAA receptor and positive modulators of the NMDA and sigma-1 receptor. There is growing evidence about the interactions between neuroactive steroids and the serotonergic system. DHEA and ALLO are believed to modulate the activity of the serotonergic neurons in the dorsal raphe nucleus, either through their direct action on these neurons or in combination with some serotonin receptor inhibitors [73,77]. Sexual steroids play an important role in the development, growth, maturation and differentiation of the Central (CNS) and Peripheral Nervous System (PNS) [75]. Moreover, estrogen and progesterone affect neuronal plasticity differently in males and females regarding changes in structure and function of neurons in different regions of the brain. SNs, namely 17β-estradiol (E2) and 5α-dehydrotestosterone (DHT), are synthesized in the hippocampus and provide some sex-specific circuit modifications, i.e., a different modification in the number of excitatory spine synapses [78]. In general, hippocampal neurons synthesize sex steroids de novo from cholesterol, since the brain is equipped with all the enzymes required for the synthesis of estradiol and testosterone, the end products of sex steroidogenesis. Locally, estradiol and testosterone maintain synaptic transmission and synaptic connectivity [79]. Remarkably, the NSs estradiol is effective in females, but not in males, and vice versa DHT is effective in males, but not in females [80]. In fact, the inhibition of estradiol synthesis in females and DHT in males causes synapse loss, LTP impairment, and synaptic protein downregulation [80]. Recently, Brökling et al. [81] showed that the upregulation of the immediate hippocampus early gene Arc/Arg3.1 is related to sex steroids in a sex-dependent manner. The cytoskeleton protein Arc/Arg3.1 is involved in synaptic plasticity, and it is crucial for long-term memory function. The authors found that E2 upregulates Arc/Arg3.1 in female, but not in male hippocampal neurons. Meanwhile, testosterone upregulates Arc/Arg3.1 in male hippocampal neurons but not in female. The block of Arc/Arg3.1 protein expression in knockout mice impairs LTP without affecting short-term memory. It could be assumed that sex steroids influence hippocampal plasticity and memory by the challenging of LTP, Arc/Arg3.1 utilization and other mechanisms that lead to spine formation. A study conducted on mice found that treatment with letrozole reduced spine synapse density in female but not in male animals [82]. 

Lu et al. [83] also carried out a study on forebrain-specific aromatase knock-out mice and showed an upregulation of estradiol receptors (ER) ER*β* and a downregulation of ER*α* in the hippocampus, and reduced the number of mushroom spines with impairment of LTP. 

Fester et al. [84] found a strong sex-dependency in estradiol-induced synaptic plasticity. In the female hippocampus, either in vivo or in hippocampal tissue which originate from female animals, locally synthesized estradiol sustains hippocampal connectivity and LTP. On the other hand, testosterone and dihydrotestosterone are responsible for the sex steroid-induced synaptic plasticity specifically in males, either for the density of mushroom spines or of spine synapses. 

Regarding hippocampal cultures of male animals, the authors found that the sex-dependent development of neurons took place in response to sex NSs, either directly encouraged by sex chromosomes or indirectly by fetal testosterone (T) secretion during the perinatal period. In fact, the authors showed that, in vitro, the synaptic plasticity as well as LTP and the stability of synapse density in the hippocampus is controlled by E2 in female and testosterone in male animals, respectively, in a sex-specific manner, and that sex steroids act in a paracrine manner [85]. 

Synapse loss in females was paralleled by the downregulation of NR1 subunits of NMDA receptors after removing the main source of estradiol in females [86]. It was demonstrated that estradiol, in females, is important for synaptogenesis, rather than estradiol from peripheral sources such as the ovaries [87]. The loss of synapses in females was found only in the hippocampus and not in other regions of the brain or cerebellum [88]. Brandt et al. showed that synapse density in the hippocampus is controlled by sex NSs in a sex-specific manner. Therefore, the preservation of synapses and mature (mushroom) spines is mediated by T and its metabolite DHT (by 5 *α*-reductase) in the hippocampus of males, E2 derived from the conversion by aromatase of hippocampus-derived T, in females [85]. 

Data on estradiol that regulated cognitive abilities come from studies of estradiol replacement therapy in pre- and postmenopausal women. 

## 5. Sexual Neurosteroids and Neurologic Disorders

### 5.1. Epilepsy 

NSs have been related to the physiological control of seizure susceptibility only in patients with epilepsy, in specific seizures such as catamenial epilepsy, stress and temporal lobe epilepsy or alcohol withdrawal [11] 

Physiologically, progesterone positively influences the seizure threshold by its conversion to ALLO that, as reported previously, strongly and positively modulates tonic GABAergic inhibition. Therefore, the pathophysiology of catamenial epilepsy (CE), a cyclical exacerbation of seizure during the menstrual cycle, has been related to the sudden decrease of hematic progesterone before menstruation in a high proportion of women with drug refractory epilepsy [89]. Antiepileptic therapy with the oral contraceptive agent medroxyprogesterone has been used effectively because of its capacity to increase the synthesis of ALLO.

Trivisano et al. [90] showed a significant reduction in cortisol and PS levels in postpubertal CE girls and low levels of ALLO, and meanwhile no differences were found in prepubertal CE girls [91]. 

Therefore, these results encouraged trials on ganaxolone, a synthetic analogue of ALLO, for its effect on epilepsy [89,92,93]. 

Even the seizure incidence, severity, and antiseizure medication (ASM) efficacy are gender dependent, especially in signaling pathways that determine network excitability. It has been shown that the electroneutral cation-chloride cotransporters (CCCs) of the SLC12A gene family have a strong influence over the electrical response to the inhibitory neurotransmitter GABA. CCCs dysfunction has been linked to seizures during early postnatal development, epileptogenesis, and refractoriness to ASMs in a sex-dependent manner [94]. However, better knowledge on the pathophysiology of sex-dependent signaling in epilepsy is fundamental to reduce long-term epilepsy comorbidities, such as lower scores on neuropsychological tests and attention-deficit hyperactivity disorder symptoms in children [94]. 

### 5.2. Headaches

Migrainous auras and mechanisms of nociceptive sensitization have been related to a disproportion between excitatory and inhibitory neurotransmission, and consequently to a hyperactivation of N-methyl-d-aspartate receptors [95]. Owing to their mechanisms of action, NSs impact the pathophysiology of migraines by altering the peripheral production of progesterone and other NSs precursors (e.g., the pre-menstrual period) by the changes in γ-aminobutyric acid mediated neurotransmission and abnormalities of neuronal excitability [96]. 

Koverech et al. [97] found that fluctuations of NSs levels are associated with chronic headache disorders. The authors demonstrated an increase in AP levels (that behave as a positive allosteric modulator) of both synaptic and extrasynaptic γ-aminobutyric acid A and reduced DHEA in chronic migraine (CM), and reduced DHEAS only in CM patients (DHEA and DHEAS behave as weak negative allosteric modulators or γ-aminobutyric acid A receptors). 

During migraine attacks, patients affected by CM + MO, showed reduced AP levels and AP/EAP ratio (EAP is an NS competitive antagonist at the AP site of γ-aminobutyric acid A receptors) and increased in DHEAS levels [98]. Whether the NS levels influence or are influenced by the attacks is still under debate [97]. 

On the other hand, in cluster headaches (CH) the AP levels were reduced as well as testosterone [98], giving the message of different and more severe symptoms with respect to EM and CM, but the same pathophysiological mechanisms of increased neuronal excitability. Whether the AP and T levels depend on the reduction of progesterone is still under investigation, because of the prevalence of CH in males. 

### 5.3. Brain Injury and Neurosteroids

Neurosteroids have been studied in traumatic injury, cerebral ischemia, multiple sclerosis, and other neuropsychiatric disorders, This date highlighted an exertion of trophic and neuroprotective effects of progesterone, DHEA, testosterone, and estradiol, in in vitro studies.

A recent case–control study on the gender differences of NS concentrations in schizophrenic patients showed DHEA-S and pregnenolone levels as significantly higher in males than in females, and a positive correlation with age of onset and negative correlation with the duration of illness in schizophrenic males, while pregnenolone serum levels demonstrated a positive correlation with the severity of anxiety, depression and cognitive impairment [99]. 

Reduced CSF levels of ALLO have been associated with depression. Therapy with fluoxetine and DHEA alleviates depressive symptoms. 

Even alcohol has been related to impaired levels of NSs mediated by modifications in GABA_A_ receptor subunit expression, suggesting that an imbalance between excitatory and inhibitory signaling in alcohol use disorders (AUD) and therapy with ALLO have shown positive results for the treatment of alcohol dependence and withdrawal [100].

Studies on multiple sclerosis (MS) have shown that NSs exert local effects on glial and neuronal tissue. Specifically, progesterone products act as promyelinating features continuously during the extensive process of myelin preservation in the adult human brain [101]. 

It has been shown that two enzymes that convert progesterone to its promyelinating metabolites (DHP and ALLOPREG), the 5-alpha reductase (5-ARD)-3-alpha-hydroxy-steroid dehydrogenase (3-alpha HSD), are in the white matter, indicating a function in central myelination [102,103,104]. 

These two enzymes regulate the intracellular concentration of steroid hormones in the myelinating cells in the central and peripheral nervous systems. The NS concentrations were equal in men and women, with variations in the female brain during the menstrual cycle [105,106]. 

Leitner et al. [101] showed that a deficit of promyelinating NSs in the nervous system plays a decisive role in the pathogenesis of multiple sclerosis. They showed that a deficiency of ALLOPREG and DHP, that regulate the neuronal-glial crosstalk necessary for myelin maintenance, is related to demyelination and reduction and impaired myelin protein composition in the whole white matter. 

Therefore, the etiology of MS is a dysregulation of NS synthesis that causes an impairment of myelin maintenance as well as remyelination and increased vulnerability of the myelin sheath, phagocytosis of myelin debris, and processing of myelin antigen by microglia, with the subsequent formation of focal demyelinating lesions over months and years. Inadequate remyelination fails in most MS lesions and leads to a persistence of the disease. Then, the authors concluded that a new therapeutic approach based on hormonal replacement with DHP alone or in combination with ALLOPREG may be helpful in MS research [101]. 

## 6. Neurosteroids and Cognition 

NSs are believed to have an important role in cognitive function. Some studies indicated a positive function of estradiol in memory [107,108], with a reduction of the risk of dementia [109]. In males, it was shown that spatial memory is improved in response to androgen treatment [92]. In tests for cognitive impairment, the pharmacological inhibition of estradiol synthesis did not affect cognitive performance in boys. DHT reverses the alteration of synaptic transmission following gonadectomy in male mice [110]. Recently, androgens have been shown to be important in achieving optimal results in working memory tests in male animals [111]. Studies conducted on female gender showed that, after menopause, when levels of estradiol decreased, cognitive impairment could increase significantly, while those who maintained higher levels of estradiol showed better performance in executive functions than females that not received estradiol in menopause [112].

Duarte-Guterman et al. [113] highlighted that estrogen type, treatment duration, dose, and treatment timing influence the effects of estrogen on spatial memory, on cognition, and in neurogenesis. Therefore, the results of these studies showed that the beneficial effects of estradiol on cognitive function could be a method to prevent cognitive impairment and, consequently, the neurodegenerative process, as well as improving synaptic plasticity in subjects at an early stage of the disease [114]. 

Some studies showed that therapies based on estradiol could lead to a slower cognitive decline and may represent a protective factor for the development of Alzheimer’s Disease (AD) [115]. However, different studies found controversial results about the effect of estrogen in AD patients [116,117]. A clinical study revealed that estrogen replacement therapy for one year had no effect on the progression of AD in females [118]. Postmortem studies conducted in AD patients showed a significant reduction in estrogen level and in the maturation of adult neurons in the dentate gyrus compared to healthy subjects [119]. However, the administration of estradiol may provide gender-specific prevention and therapeutic approaches for AD [63,64,65]. In addition, several studies demonstrated the correlation between the effects of estradiol on neurogenesis in the hippocampus and its therapeutic effects to improve spatial memory in AD [63,64]. A significant increase in hippocampal neurogenesis was associated with an improvement of cognitive deficits after estrogen therapy during the early stage of AD in the Aβ1-42 mouse model of AD [83]. It is important to understand the mechanism of neurogenesis in response to estradiol to promote novel cognitive and physical treatments to improve neurogenesis in AD, and consequently increase synaptic plasticity [65]. The molecular mechanisms of the interaction between estradiol and the loss of synapses are unclear to the scientific community. More experiments focused on this mechanism could be useful to design an appropriate therapeutic intervention that may reverse or delay the progress of neurodegeneration in AD. Most of the available data are collected by studies on animal models, caused by the limitations of adult hippocampal neurogenesis in humans. NSs have also been related to Niemann–Pick type C (NP-C), characterized by impaired cholesterol trafficking and reduced levels of ALLO. The brains of mice with NP-C present widespread CNS demyelination, Purkinje cell loss, and motor impairment. Treatment with the neonatal administration of ALLO increased cell survival, delayed the appearance of neurologic symptoms, and approximately doubled the lifespan of the mice [120]. 

## 7. Neurosteroids and Rehabilitation 

To date, the scientific community has questions about the possibility of modulating the hormonal system and how to promote well-being and improve cognitive and physical performance. The regulation of physical and mental health also depends on the maintenance of skeletal muscle mass, the physical activity being influenced by hormones such as testosterone, E2, growth hormone (GH), and insulin-like growth factor (IGF). The importance of these hormones for the regulation of skeletal muscle mass is known, but their interaction with the processes controlling muscle mass remain unclear [121]. Few studies have demonstrated the link between physical training and hormonal response, although resistance exercise elicited a significant hormonal response. It has been seen that physical activity at a certain level influenced the hormonal response. Anabolic hormones, such as testosterone and GH, are elevated during the 15 to 30 min following resistance exercise if an adequate stimulus is present. Indeed, moderate or high intensity training that stresses a large muscle mass tends to produce higher acute hormones (testosterone, GH, and the catabolic hormone cortisol) than low-intensity training [122]. 

Razzak et al. [123] investigated the effects of aerobic and anaerobic exercises on postmenopausal women. The study concluded that 12 weeks of anaerobic exercise programs improved women’s E2 levels better than aerobic exercise, as a protective factor not only to maintain physical well-being but consequently cognitive function, by regulating adult neurogenesis of the hippocampus and synaptic plasticity, learning skills and memory [114]. 

Cognitive and motor rehabilitation is itself a non-invasive neuromodulation tool that creates favorable conditions to promote changes in nerve impulse transmission for therapeutic purposes, thus supporting neuroplasticity. According to the literature, this phenomenon is most encouraging with the use of invasive neuromodulation tools (such as Transcranial Magnetic Stimulation and transcranial Direct Current Stimulation), and we could hypothesize that the combination with NS therapy enhances the neuroplasticity effect [58,113,123]. Indeed, it was found that SN, such as E2, influence spatial memory and other cognitive functions, and this depends on the type of E2 therapy, the duration of treatment and the dose. On the other hand, Mohammadi at al. [123] described the importance of using Transcranial Magnetic Stimulation to induce neuromodulation effects. In previous studies, it has been found that E2 administration in postmenopausal women allowed for improving cognitive performance [112] and that the combination with aerobic motor exercise would enhance this effect. For this reason, we could hypothesize that the paired association between neuromodulatory techniques, motor rehabilitation and NSs treatment could provide a boosting effect in order to promote neuroplasticity.

## 8. Future Perspective

NSs endogenously regulate neural excitability by the allosteric potentiation of GABA-A receptors. Although NS biosynthesis in the brain is well known, the controlling molecular mechanisms must be enhanced. Current neuroscience studies on neuroactive steroids have demonstrated their efficacy in some neurological conditions, although future investigations are required to improve information about the NSs impact on the human brain. NS research continues to provide new insights into the mechanisms of action that may influence therapeutic intervention. Further studies are needed to improve the hormonal impact on brain function, especially in females, and the specific NSs role in gender-specific brain conditions. To this end, there is renewed interest in synthetic NS analogs as promising therapeutic agents in clinical practice to manage several neurological disorders. Indeed, there are currently at least some compounds in clinical trials for epilepsy, traumatic brain injury, status epilepticus, and Fragile X syndrome, as well as unmet neuropsychiatric disorders.

Moreover, NSs may be positively used to improve neurodevelopmental disorders. In fact, alterations of ALLO levels during maturation could partly explain the inter-individual differences shown by adolescents in response to novelty (exploration) and in the sensorimotor gating and prepulse/impulse inhibition in adults. 

These data underline that the spreading of a set of interneurons is in response to neonatal NS exposure and the importance of their mechanisms’ maturation in the brain related to emotionality and the responses to environmental stressors in adulthood. Longitudinal studies in these patients are necessary to establish whether the use of NSs may change the natural history of these disabling disorders. Finally, concerning the rehabilitation field, it could be investigated as to how and to what extent NSs integrating with virtual reality and other innovative technologies may boost neuralplasticity, and therefore the motor and cognitive outcomes of patients with neurological disorders. 

## 9. Conclusions

In conclusion, NSs are potent endogenous neuromodulators with rapid actions in the brain via non-genomic mechanisms. Indeed, they modulate different neurotransmitter pathways and act in a different way in the female and male brain. As they have a pivotal role in neurogenesis, the absence or reduced concentrations of NSs during development may be associated with various neurodevelopmental and neuropsychiatric disorders such as anxiety disorders, schizophrenia, epilepsy, or AD. The effect of NS compounds should be investigated in future studies in order to establish if and to what extent they may change the course of neuropsychiatric disorders. 

## Figures and Tables

**Figure 1 ijms-24-04758-f001:**
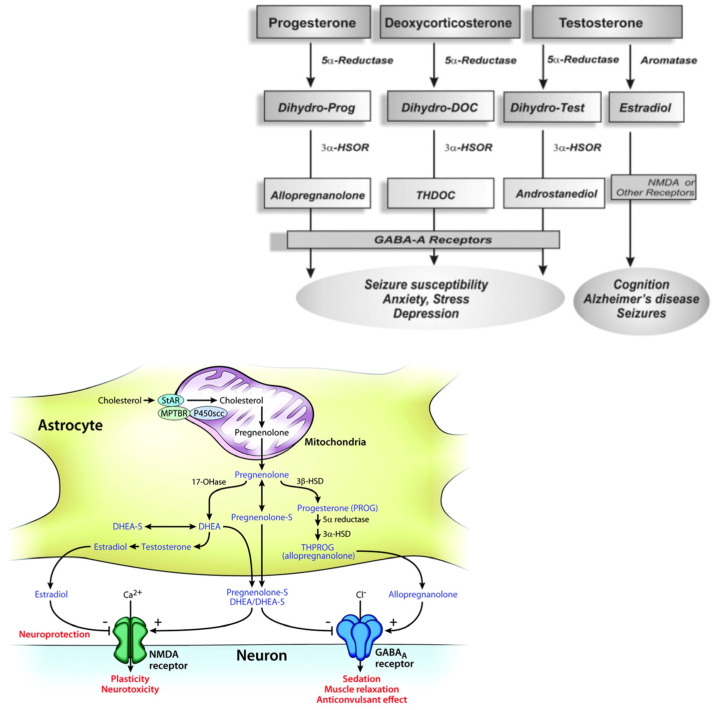
Biosynthetic pathways for steroidogenesis. The conversion of progesteron, deoxycoricosteron and testosterone in allopregnenolone, THDOC and androstanediol by the enzyme 5α-Reductase, aromatase and 3α-HSOR, impacting on brain functions (molecular view; schematic illustration); the GABA-A receptors composed of α, β and γ or δ subunits structure that form chloride ion channel.

## Data Availability

Not applicable.

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
