# Peer review of "Gender and Neurosteroids: Implications for Brain Function, Neuroplasticity and Rehabilitation"

_ijms, 2023, doi:10.3390/ijms24054758_

Round 1

Reviewer 1 Report

In this manuscript the authors reviewed Neurosteroids implications for brain function, neuroplasticity and rehabilitation for both females and males.

This review in principle is interesting; unfortunately, this manuscript needs very manyimprovements and corrections before publishing may be possible.

General points:

Please add a list of abbreviations before References section to your manuscript.

The use of abbraviations in teh manuscript is inconsistent and confusing. Introduce it once and then uns it throughout the manuscript.

The language definitely needs a correction by a native speaker.

The whole manuscript must be rearranged to become more strictly. In the introduction, many facts are found several times, partly confusing and contardictory.

One of my big problems following the manuscript: have all neurosteroids the same actions and mechanisms of actions? I cannot see a differnatione of the various molecules in the manuscript.

Special points:

Please add a Future perspectives section to your manuscript.

Introduction

The arrangement of the introduction section is somewhat confusing. Please rearrange the parapgraph in a more coherent way as: what are NS, were do they come from, were are they or their precursors synthesized, were are they found, what are they suggested to do, ...

Lines 38-41: please add multiple references at the end of each of these sentences.

Lines 54-60: These sentences are unclear and contradictory.

Fig 1: The arragement of arrows and lines between the various fields is not consistent: what does an arrow mean, what a line? 

Where are A-C -mentioned in the legend - are found in the Fig.1?

Fig.2: The Fig is confusing and gives no sense to me: what is the meaning of the different fonts or bold formations? At least, there are no pathways seen. In the present form it is useless.

Lines 84-86: please add multiple references at the end of this sentence.

Lines 88-89: please correct the multiple dots at the end of this sentence.

Lines 91-97: please make a more compact and clear description without repetitions.

Lines 98-100: this was already mentionned.

Lines 101-103: please add multiple references at the end of this sentence.

Lines 104-107: please add multiple references at the end of this sentence.

Lines 108-112: please add multiple references at the end of this sentence.

Lines 112-114: what does this sentence mean?

Lines 120-124: please add multiple references at the end of this sentence.

Line 144-146: this sentence gives no sense for me.

Line 176: what is SN? In this and the following parts of the ms.

Lnes 176-185: this is confusing - what do you wnat to state?

Lines 179-182: please add multiple references at the end of this sentence.

Line 188; what means .. reached ..

Lines 205-212: please add multiple references at the end of these sentence.

Lines 214-220: please add multiple references at the end of each of these sentences.

Line 229: what is the meaning of this subheading, when no further subheading follows?

Lines 235-241: please add multiple references at the end of each of these sentences.

Lines 303-308: please add multiple references at the end of each of these sentences.

Lines 344-346: please describe more exactly all these studies.

Author Response

Reviewer 1 comments and response of author:

In this manuscript the authors reviewed Neurosteroids implications for brain function, neuroplasticity and rehabilitation for both females and males.

This review in principle is interesting; unfortunately, this manuscript needs very manyimprovements and corrections before publishing may be possible.

General points:

Please add a list of abbreviations before References section to your manuscript.

Done

The use of abbraviations in teh manuscript is inconsistent and confusing. Introduce it once and then uns it throughout the manuscript.

Done

The language definitely needs a correction by a native speaker.

Done

The whole manuscript must be rearranged to become more strictly. In the introduction, many facts are found several times, partly confusing and contardictory.

One of my big problems following the manuscript: have all neurosteroids the same actions and mechanisms of actions? I cannot see a differnatione of the various molecules in the manuscript.

Thank you for your observation. We clarified some points in introduction section and we merged the biosynthesis paragraph in the introduction. We described better the mechanism of action in the dedicated section.

Special points:

Please add a Future perspectives section to your manuscript.

Done

Introduction

The arrangement of the introduction section is somewhat confusing. Please rearrange the parapgraph in a more coherent way as: what are NS, were do they come from, were are they or their precursors synthesized, were are they found, what are they suggested to do, ...

Thank you for your observation. We clarified some points in introduction section and we merged the biosynthesis paragraph in the introduction. We described better the mechanism of action in the dedicated section.

Lines 38-41: please add multiple references at the end of each of these sentences.

Done

Lines 54-60: These sentences are unclear and contradictory.

We deleted the sentence.

Fig 1: The arragement of arrows and lines between the various fields is not consistent: what does an arrow mean, what a line? 

Where are A-C -mentioned in the legend - are found in the Fig.1?

We clarified the legend of the figure.

Fig.2: The Fig is confusing and gives no sense to me: what is the meaning of the different fonts or bold formations? At least, there are no pathways seen. In the present form it is useless.

We deleted the figure

Lines 84-86: please add multiple references at the end of this sentence.

Done

Lines 88-89: please correct the multiple dots at the end of this sentence.

Done

Lines 91-97: please make a more compact and clear description without repetitions.

Done

Lines 98-100: this was already mentionned.

We deleted the sentence

Lines 101-103: please add multiple references at the end of this sentence.

Done

Lines 104-107: please add multiple references at the end of this sentence.

Done

Lines 108-112: please add multiple references at the end of this sentence.

Done

Lines 112-114: what does this sentence mean?

We deleted this sentence.

Lines 120-124: please add multiple references at the end of this sentence.

Done

Line 144-146: this sentence gives no sense for me.

I deleted the sentence

Line 176: what is SN? In this and the following parts of the ms.

The abbreviation is Sexual Neurosteorids. I specified.

Lnes 176-185: this is confusing - what do you wnat to state?

We deleted the sentence.

Lines 179-182: please add multiple references at the end of this sentence.

Done

Line 188; what means .. reached ..

I deleted

Lines 205-212: please add multiple references at the end of these sentence.

Done

Lines 214-220: please add multiple references at the end of each of these sentences.

Done

Line 229: what is the meaning of this subheading, when no further subheading follows?

Deleted

Lines 235-241: please add multiple references at the end of each of these sentences.

Done

Lines 303-308: please add multiple references at the end of each of these sentences.

Done

Lines 344-346: please describe more exactly all these studies.

Done

All corrections are revised in main test with “track changes”.

I realized in the manuscript that there were  different errors in email association with authors of the manuscript. Therefore, I felt it necessary to insert them correctly. The changes are highlighted in track changes in the manuscript.

Reviewer 2 Report

The manuscript “Gender And Neurosteroids: Implications for Brain Function, Neuroplasticity and Rehabilitation” by Raciti et al is a review article which describes the roles of neurosteroids in brain function, neuroplasticity and rehabilitation. Generally, the subject is of interest and scientifically sound and contains essential contents. This topic is also of importance for understanding the roles of neurosteroids. The manuscript has been well organized and written. However, I have some concerns on the paper.

Although the title indicates that this review describes gender specific roles of neurosteroids, I think that the topics about gender differences are less and not thoroughly described.

Several grammatical errors are found as described below.

P3, lines 88-89: “Nevertheless, the controlling mechanisms of the neurosteroid 88 biosynthesis in the brain is still unclear.”

P3, lines 94-95: The effects of neurosteroids on modulation of brain excitability is the result of the 94 contact with neuronal membrane receptors.

Author Response

Reviewer 2 comments and response of author:

The manuscript “Gender And Neurosteroids: Implications for Brain Function, Neuroplasticity and Rehabilitation” by Raciti et al is a review article which describes the roles of neurosteroids in brain function, neuroplasticity and rehabilitation. Generally, the subject is of interest and scientifically sound and contains essential contents. This topic is also of importance for understanding the roles of neurosteroids. The manuscript has been well organized and written. However, I have some concerns on the paper.

Although the title indicates that this review describes gender specific roles of neurosteroids, I think that the topics about gender differences are less and not thoroughly described.

Response author: Dear reviewer, thank you to consider my manuscript. As you suggested, I added other references about gender differences in gender differences of action. other differences in geneder are described in Neurosteroids and Cognition and Neurosteroids and Rehabilitation sections. Moreover, We have added more information on gender-specific effect as well as the role of Ns in different neuropsychiatric disorders.

Several grammatical errors are found as described below.

P3, lines 88-89: “Nevertheless, the controlling mechanisms of the neurosteroid 88 biosynthesis in the brain is still unclear.”

P3, lines 94-95: The effects of neurosteroids on modulation of brain excitability is the result of the 94 contact with neuronal membrane receptors.

Response Author: I corrected the grammatical errors as you suggested. 

All corrections are revised in main test with “track changes”.

I realized in the manuscript that there were  different errors in email association with authors of the manuscript. Therefore, I felt it necessary to insert them correctly. The changes are highlighted in track changes in the manuscript.

Reviewer 3 Report

IJMS

COMMENTS TO THE EDITORS AND THE AUTHORS

ijms-2225139: “Gender And Neurosteroids: Implications for Brain Function, Neuroplasticity and Rehabilitation”
Dear the Editors and the Authors,

 Please find enclosed the comments for the above-mentioned manuscript.

 A SUMMARY OF THE CONTENT. The authors described some of the roles of the neurosteroids in the regulation of  the neural excitability by allosteric potentiation of GABA-A receptors. They emphasized that the controlling molecular mechanisms related to the neurosteroids biosynthesis are not precisely stated and that future investigations are required to improve information about the neurosteroids on the human brain.

 THE OVERALL OPINION OF THE MANUSCRIPT

The strengths. 

The manuscript is within the scope of the journal and addresses the important question. The text is very easy to follow. The figures very clearly present the content of the text.

The limitations.

The title is not fully supported by the text; some of the descriptions are not precise; the citation of the original and important recent advances in the field are missing; the discussion of the sex-dependent observation is missing; the references are not consistently cited.

SUGGESTIONS

(1) To keep the title, please discuss the sex-dependent differences in the responses through whole text/chapters of the manuscript. Namely, the author dedicated only one small chapter to “Gender differences of action“. That is one point. The other point is sex-dependent responses to neurosteroids in different cells/tissues and at different age.

(2) Please state the aim of the review in the abstract and in the introduction. It will increase “visibility” of your manuscript. 

(3) Please avoid citation of the reference in the abstract since “71” does not mean anything to the readers.

(4) Please describe and the discuss the original and the important recent advances in the field focusing on the subject of the study instead of the review articles.

(5) Please precisely describe the source of the results (type and sex of the cells, tissue etc.).

(6) Please describe and discuss the sex-dependent differences in the responses.

(7) Please describe and discuss the age-related differences in the responses.

(8) Please consider making clear conclusions related to the text described in the manuscript.

(9) Please consider adding one separate paragraph describing the future perspectives.

(10) Please make consistent citation of the references. Namely, the name of the some of the journals are cited in full, some are abbreviated, some with capital first letter, some not, etc. For some references authors provided “doi”, for some not.

Accordingly, minor revision is required.

I would greatly appreciate if you will contact me if you find something in my comments is missing/unclear/incorrect.

Good luck and all the best J

Author Response

Reviewer 3

 THE OVERALL OPINION OF THE MANUSCRIPT

The strengths. 

The manuscript is within the scope of the journal and addresses the important question. The text is very easy to follow. The figures very clearly present the content of the text.

  • Thank you for the positive evaluation.

The limitations.

The title is not fully supported by the text; some of the descriptions are not precise; the citation of the original and important recent advances in the field are missing; the discussion of the sex-dependent observation is missing; the references are not consistently cited.

All these concerns have been addressed as better specified in the point by point response.

(1) To keep the title, please discuss the sex-dependent differences in the responses through whole text/chapters of the manuscript. Namely, the author dedicated only one small chapter to “Gender differences of action“. That is one point. The other point is sex-dependent responses to neurosteroids in different cells/tissues and at different age.

Thank you for your interesting observation. We added information about sex-dependent responses to neurosteroids in the text (reported in bold type).

(2) Please state the aim of the review in the abstract and in the introduction. It will increase “visibility” of your manuscript.

We added the aim of the review in the abstract and introduction sections as follow:

The aim of this review is to investigate the mechanism of action of the neurosteroids, their sex-dependent differences within brain function, and their role in neuroplasticity and rehabilitation”

(3) Please avoid citation of the reference in the abstract since “71” does not mean anything to the readers.

We delete the reference reported in the abstract.

(4) Please describe and the discuss the original and the important recent advances in the field focusing on the subject of the study instead of the review articles.

Thank you for your suggestion. We improve the text content discussing the researchers found on neurosteroids, avoid reviews. Moreover, we added researchers on neurosteroids and neurological diseases.

(5) Please precisely describe the source of the results (type and sex of the cells, tissue etc.).

Done

(6) Please describe and discuss the sex-dependent differences in the responses.

Done

(7) Please describe and discuss the age-related differences in the responses.

Done, as suggested.

Indeed, As previously reported, we added and discuss scientists studies on sex-dependent responses to neurosteroids through the text.

(8) Please consider making clear conclusions related to the text described in the manuscript.

Conclusions have been modified, as suggested.

(9) Please consider adding one separate paragraph describing the future perspectives.

Done, as suggested

Thank you for your observation. We add the paragraph as suggested.

(10) Please make consistent citation of the references. Namely, the name of the some of the journals are cited in full, some are abbreviated, some with capital first letter, some not, etc. For some references authors provided “doi”, for some not.

Done

Thank you for your comment. We revised all the references.

I realized in the manuscript that there were different errors in email association with authors of the manuscript. Therefore, I felt it necessary to insert them correctly. The changes are highlighted in track changes in the manuscript.

Round 2

Reviewer 1 Report

The authors massively improved the manuscript and specified the criticized points. 

There are still some shortcommings: English language, typing errors, puctuation errors, and errors in formating the manuscript.

I think, if these things get fixed, the manuscript is a useful manuscript.

Author Response

thank you for your suggestion. 

we revised the manuscript as you suggested. 
